# Prediction of five-year mortality after COPD diagnosis using primary care records

Steven J. Kiddle[1,2]*, Hannah R. Whittaker[2], Shaun R. Seaman[1], Jennifer K. Quint[2]*

1 MRC Biostatistics Unit, University of Cambridge, Cambridge, United Kingdom, 2 National Heart and Lung Institute, Imperial College London, London, United Kingdom

* steven.kiddle@mrc-bsu.cam.ac.uk (SJK); j.quint@imperial.ac.uk (JKQ)

**Data Availability Statement:** The data used in our study originates from UK General Practice health records using the Vision software, and is provided in anonymised form by the Clinical Practice Research Datalink (CPRD, https://www.cprd.com/)

## Abstract

Accurate prognosis information after a diagnosis of chronic obstructive pulmonary disease (COPD) would facilitate earlier and better informed decisions about the use of prevention strategies and advanced care plans. We therefore aimed to develop and validate an accurate prognosis model for incident COPD cases using only information present in general practitioner (GP) records at the point of diagnosis. Incident COPD patients between 2004–2012 over the age of 35 were studied using records from 396 general practices in England. We developed a model to predict all-cause five-year mortality at the point of COPD diagnosis, using 47,964 English patients. Our model uses age, gender, smoking status, body mass index, forced expiratory volume in 1-second (FEV1) % predicted and 16 co-morbidities (the same number as the Charlson Co-morbidity Index). The performance of our chosen model was validated in all countries of the UK (N = 48,304). Our model performed well, and performed consistently in validation data. The validation area under the curves in each country varied between 0.783–0.809 and the calibration slopes between 0.911–1.04. Our model performed better in this context than models based on the Charlson Co-morbidity Index or Cambridge Multimorbidity Score. We have developed and validated a model that outperforms general multimorbidity scores at predicting five-year mortality after COPD diagnosis. Our model includes only data routinely collected before COPD diagnosis, allowing it to be readily translated into clinical practice, and has been made available through an online risk calculator (https://skiddle.shinyapps.io/incidentcopdsurvival/).

## Introduction

Chronic obstructive pulmonary disease (COPD) is the fifth highest cause of death in the United Kingdom (UK) [1]. One of the goals of COPD diagnosis and assessment is to provide information about the risk of future events such as death in order to make informed decisions about the use of primary and secondary prevention strategies, and advanced care plans [2]. However, existing prognosis models focus on prevalent COPD, rather than incident cases, meaning that they depend on variables which are often not recorded in GP records at the time of COPD diagnosis. Additionally, external validation of these models appears to be rare, and when performed have resulted in inconsistent findings [3–5].

to approved researchers for approved projects under strict conditions as assessed by their Independent Scientific Advisory Committee (isac@cprd.com) which holds broad ethics approval and is responsible for ensuring projects are covered by this. CPRD are the only entity legally allowed to share this data which although anonymised has the potential for reidentification in some cases. We attach our ISAC approved protocol and make our analysis scripts open source in order to help other researchers take the steps necessary to get approval from ISAC to reproduce our findings.

**Funding:** SJK is supported by a MRC Career Development Award (MR/P021573/1). SRS is supported by MRC Programme Grant (MC_UU_00002/10). The funders had no role in the decision to publish.

**Competing interests:** Dr. Kiddle reports grants from Medical Research Council, during the conduct of the study; personal fees from Roche Diagnostics and DIADEM, outside the submitted work. After completing this work, but before manuscript submission Dr. Kiddle became an employee of AstraZeneca. Ms. Whittaker reports grants from GlaxoSmithKline, during the conduct of the study. Dr. Seaman has nothing to disclose. Dr. Quint reports grants from MRC, grants from The Health Foundation, grants from BLF, grants and personal fees from GSK, grants and personal fees from BI, grants and personal fees from Insmed, grants and personal fees from AZ, personal fees from Chiesi, personal fees from Teva, outside the submitted work. This does not alter our adherence to PLOS ONE policies on sharing data and materials.

A key predictor of mortality is the presence of co-morbidities, as demonstrated by the Charlson co-morbidity index, which takes into account age and the presence of 16 diseases [6]. More recently, Rupert Payne et al., (under review) have developed the Cambridge multimorbidity score. This uses data on the presence or absence of 20 diseases, and performs slightly better than the Charlson co-morbidity index. The deaths of up to two thirds of COPD patients are thought to be due to co-morbidities [7–10]. However, existing COPD prognosis models that include co-morbidities have been developed either in small cohorts, or in populations unrepresentative of general practice or with small lists of co-morbidities [8,11,12]. An exception to this was developed using data on 59,990 patients from UK general practice, but again this focused on prevalent cases and performed worse in a validation cohort [13].

In this study we sought to develop and validate GP-record-based (i.e. not claims based) models predicting survival for incident COPD patients, focusing on longer-term survival (5-year). We aimed to produce a model that could be implemented in a user-friendly website. Importantly, we sought to make predictions based on data available at or before the point of diagnosis, often before many of the variables used in COPD prognosis models have been logged within GP records, such as dyspnea and FEV1% predicted (required for the BARC model). Our aim was to provide accurate predictions of survival for individuals based on their baseline characteristics.

## Materials and methods

### Data source

Data from Clinical Practice Research Datalink (CPRD)-GOLD (March 2017 release) were used to develop and validate the prognostic model. CPRD-GOLD, which is based on the Vision GP health record system (i.e. not a claims database), is representative of the UK population [14]. Data on mortality and socioeconomic status were collected through linkage (where available) to Office for National Statistics (last death date 19th September 2017) and Index of Multiple Deprivation 2010, which were available for approximately 60% of CPRD-GOLD practices, all of which are based in England (linkage set 15). Mortality data for patients who could not be linked were derived from CPRD-GOLD, which has been shown to be approximately correct [15]. However, vital status for CPRD-GOLD patients without linkage to ONS are not known if they have transferred out of practice before end of study. CPRD-GOLD data are available to approved researchers for approved projects (https://www.cprd.com/). The protocol for this study, which is covered by the CPRD Independent Scientific Advisory Committee ethics approval, is provided S1 File.

### Study population

All patients received their first COPD diagnosis between 1st January 2004 and 19th September 2012, determined using a previously validated algorithm using diagnostic codes alone. By contacting GPs, we have shown that this algorithm has a positive predictive value of 87% for the identification of COPD patients [16]. To be included in this study, patients were required to be 35 years or older, registered at their GP practice, and belong to a practice with up-to-standard data reporting at the time of their COPD diagnosis. We further divided the patients into two groups, such that the 'linked group' belonged to a practice that allowed linkage to Office for National Statistics and Index of Multiple Deprivation whereas the 'unlinked group' did not (Fig 1). By using only the linked group to develop our model we reduce bias due to unknown death dates for individuals in the unlinked group who transferred out of practice within 5 years. The linked group contained only patients in English practices, whereas the unlinked group contained patients from across the UK.

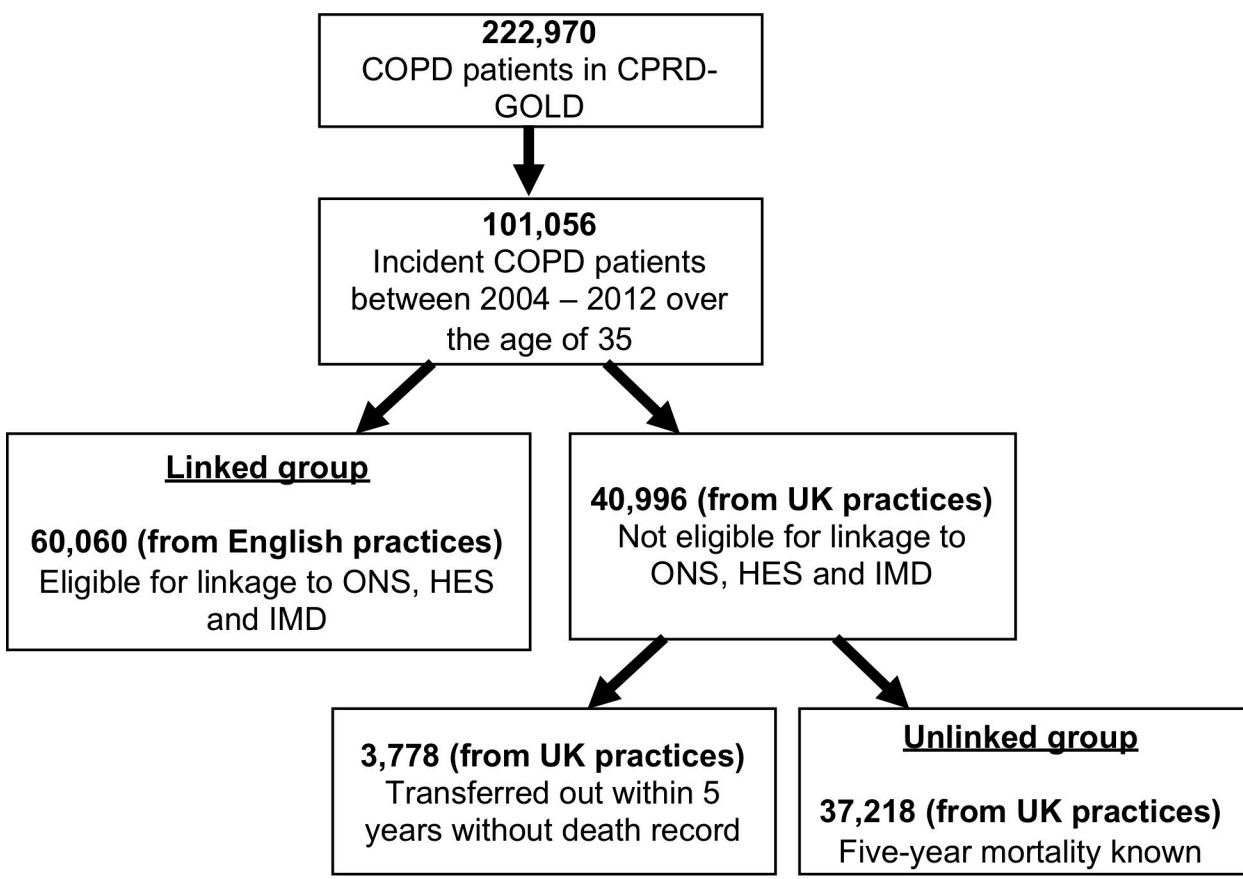

**Fig 1. Flow diagram of included patients, highlighting linked and unlinked groups.**

## Statistical software

Data analysis was performed in R 3.4.4, while data preparation was performed in both R and STATA 14. R package names are given in the following sections where appropriate (*in brackets and italics*). For transparency and reproducibility, all analysis scripts are available from https://github.com/Kiddle-group.

## Outcome and prognostic predictors

Death was defined as mortality from any cause within five years of COPD diagnosis. Prognostic predictors were divided into two categories: 'basic', and 'co-morbidities'. Basic variables were collected from visits before or on the same day as the first COPD diagnosis, and included age, gender, socioeconomic status (twentiles of Index of Multiple Deprivation), smoking status (most recently recorded: never, ex, current), body mass index (most recently recorded), body mass index not-recorded (1 = TRUE 0 = FALSE), FEV1% predicted in preceding year, and FEV1 not-recorded in preceding year (1 = TRUE, 0 = FALSE). MRC Dyspnea score was not included because of its high missingness (90%) before or on the day of COPD diagnosis.

Co-morbidities considered in this study were based on the list used in Barnett et al. [17]. To extract these conditions, we used the read and product code based definitions that have been developed by the CPRD @ Cambridge team (Rupert Payne et al., under review; https://www.phpc.cam.ac.uk/pcu/cprd_cam/codelists/v11/). For comparison we also extracted co-morbidities used in the Charlson co-morbidity index using read and product code based on liver

disease, metastatic carcinoma, dementia, hemiplegia/paraplegia from the same website, with all other codelists available on request.

Asthma was defined using an alternative codelist and approach developed for COPD patients, requiring presence of an asthma code between two–five years before COPD diagnosis, to reduce the presence of misdiagnosed patients. The Barnett co-morbidities were used to calculate the Cambridge multimorbidity score, as detailed in Payne et al., (under review).

Co-morbidities were considered as present or absent at the point of COPD diagnosis, with the exception of kidney disease which we modelled using the maximum value of eGFR from the last two measurements before COPD diagnosis. An indicator for not recording eGFR at least twice, irrespective of its value, was used (1 = eGFR tested only once or not at all, 0 = eGFR tested twice or more).

An additional co-morbidity was also added—gastro-oesophageal reflux disease recorded in the preceding year. Latest values for blood albumin, platelets and c-reactive protein as well as their corresponding not-recorded indicators were also considered in some models.

Continuous variables were median centred. Missing values in variables with a corresponding indicator of test not performed were set to the median observed value. Outliers were removed as follows: FEV1 above 5 litres, body mass index above 70 $kg/m^2$, eGFR above 200 $mL/min/1.73m^3$, c-reactive protein above 370 mg/L and albumin above 70 g/L.

## Risk prediction modelling approaches

In this study, we considered several modelling methods (logistic, survival, lasso, ridge, random forest) and sets of variables (basic, co-morbidities, co-morbidity interactions), as summarised in S1 Table. The model that we ultimately chose, which we call incident COPD prognosis (iCOPD), was based on logistic regression (*glm*) without any interaction terms, and only used 16 co-morbidities.

## Assessment of predictive ability

To avoid over-optimism about the predictive performance of any given model in the development stage, patients with linked data were randomly split into a training set of 80% of practices and a held-out test set of 20% of practices. The training set was used to fit the models and to determine which combination of model and variable set (listed in S1 Table) provided the best predictions. This was done using ten-fold cross-validation of the training set, with five replications. The predictive performances of the iCOPD model was evaluated in the held-out test set. We additionally tested iCOPD (and only this model) in the patients without linked data. To do this, we used CPRD-GOLD recorded death dates and excluded the 10% of patients whose vital status after 5-years is unknown.

The score we used to assess overall predictive accuracy was the Brier score (*rms*) which takes a value between zero and one, with lower scores indicating more accurate prediction [18]. To assess calibration we used the calibration slope (*rms*) where a slope of 1 indicates perfect calibration [18]. To assess discrimination we used the area under the curve measure (equivalent to c-index) (*rms*) which takes a value between 0 and 1, with higher scores indicating better discrimination [18]. Finally, we compared actual to predicted risk in each subgroup of the sample defined by quintiles of predicted risk (*ResourceSelection*).

## Ethics approval

The use of data from the Clinical Practice Research Datalink was approved by the CPRD-Independent Scientific Advisory Committee (16_276).

## Results

### Characteristics of the COPD population

From a total of 222,970 COPD patients in CPRD-GOLD, 60,060 patients (all in England) had linked data (from Office for National Statistics, Hospital Episode Statistics and Index of Multiple Deprivation) and 37,218 (from across the UK) did not. Patient flow is depicted in Fig 1. The median age was 68 and 67 years, and the median FEV1% predicted was 65% and 64% for the linked and unlinked patients respectively. The majority—84% in the linked and 86% in the unlinked group—had at least one of the Barnett co-morbidities by COPD diagnosis (i.e. were multimorbid), with the median number being 2. Between a quarter and a fifth—24% in the linked and 21% in the unlinked group—died within five years of their COPD diagnosis. As expected, the presence of co-morbidities was related both to age and to death. The proportion of patients whose body mass index or smoking status were not recorded was higher in those without a Barnett co-morbidity (Tables 1 and 2).

The most prevalently recorded of the Barnett co-morbidities in these patients was hypertension (38% in the linked and 37% in the unlinked group), followed by painful condition (30% in the linked and 34% in the unlinked group) and asthma (20% in the linked and 18% in the unlinked group). Only seven Barnett co-morbidities–dementia, chronic liver disease, anorexia/bulimia, Parkinson's, migraine, multiple sclerosis and learning disability–had a recorded prevalence of <1% in the linked group (S2 Table). All of these except dementia also had a recorded prevalence of <1% in the unlinked group.

The linked group was randomly split at the practice level into a training set for model development containing 47,964 COPD patients and a held-out test set of 12,096 COPD patients. Five-year mortality was 24% in both these datasets.

### Development of models within the training set

First we compared various modelling approaches, and found logistic regression to perform well (S1 Fig). Using logistic regression we wanted to develop a model, iCOPD, that uses a

**Table 1. Demographics and clinical characteristics of the linked group (those linked to hospital and ONS mortality data).**

| Variables | All eligible COPD patients (N = 60,060) | No Barnett co-morbidities (N = 9,579; 16%) | 1–2 Barnett co-morbidities (N = 24,289; 40%) | 3 or more Barnett co-morbidities (N = 26,192; 44%) |
|---|---|---|---|---|
| Female gender | 28,478 (47%) | 3,979 (42%) | 11,584 (48%) | 12,915 (49%) |
| Age, years | 68 (59–76) | 62 (54–69) | 66 (58–74) | 72 (64–79) |
| Body Mass Index, kg/m$^2$ | 26 (23–30) | 25 (22–28) | 26 (23–30) | 27 (24–31) |
| • Not recorded | 5,824 (10%) | 1,801 (19%) | 2,353 (10%) | 1,670 (6%) |
| Smoking | | | | |
| • Never smoker | 8,853 (15%) | 803 (8%) | 3,299 (14%) | 4,751 (18%) |
| • Ex smoker | 24,488 (41%) | 2,931 (31%) | 9,418 (39%) | 12,139 (46%) |
| • Current smoker | 25,632 (43%) | 5,417 (57%) | 11,154 (46%) | 9,061 (35%) |
| • Not recorded | 1,087 (2%) | 428 (4%) | 418 (2%) | 241 (1%) |
| FEV1% predicted | 65 (50–79) | 65 (49–79) | 64 (50–79) | 65 (52–79) |
| • FEV1 not recorded | 24,691 (41%) | 3,947 (41%) | 9,589 (39%) | 11,155 (43%) |
| Index of Multiple Deprivation, in twentiles | 12 (6–16) | 12 (7–16) | 11 (6–16) | 12 (7–16) |
| • Not recorded | 42 (0.1%) | 7 (0.1%) | 16 (0.1%) | 19 (0.1%) |
| Deaths within 5 years of COPD diagnosis | 14,139 (24%) | 1,389 (15%) | 4,517 (19%) | 8,223 (31%) |

Variable information are presented as either counts (percentages) or median (interquartile range).

**Table 2. Demographics and clinical characteristics of the unlinked group (those not linked to hospital and ONS mortality data).**

| Variables | All eligible COPD patients (N = 37,218) | No Barnett co-morbidities (N = 5,362; 14%) | 1–2 Barnett co-morbidities (N = 14,779; 40%) | 3 or more Barnett co-morbidities (N = 17,077; 46%) |
|---|---|---|---|---|
| **Female gender** | 18,148 (49%) | 2,403 (45%) | 7,015 (47%) | 8,730 (51%) |
| **Age, years** | 67 (59–75) | 61 (54–69) | 65 (58–73) | 70 (62–78) |
| **Body Mass Index, kg/m²** | 26 (23–30) | 25 (22–28) | 26 (23–30) | 27 (24–31) |
| • Not recorded | 3,427 (9%) | 974 (18%) | 1,467 (10%) | 986 (6%) |
| **Smoking** | | | | |
| • Never smoker | 5,304 (14%) | 501 (9%) | 1,955 (13%) | 2,848 (17%) |
| • Ex smoker | 14,109 (40%) | 1,549 (29%) | 5,250 (36%) | 7,310 (43%) |
| • Current smoker | 17,163 (46%) | 3,082 (57%) | 7,319 (50%) | 6,762 (40%) |
| • Not recorded | 642 (2%) | 230 (4%) | 255 (2%) | 157 (1%) |
| **FEV1% predicted** | 64 (50–76) | 63 (48–76) | 64 (50–76) | 64 (51–76) |
| • FEV1 not recorded | 15,808 (42%) | 2,230 (42%) | 5,932 (40%) | 7,646 (45%) |
| **Region** | | | | |
| • England | 12,994 (35%) | 1,997 (37%) | 5,375 (36%) | 5,622 (33%) |
| • Northern Ireland | 3,288 (9%) | 460 (9%) | 1,187 (8%) | 1,641 (10%) |
| • Scotland | 10,325 (28%) | 1,538 (29%) | 3,999 (27%) | 4,788 (28%) |
| • Wales | 10,611 (29%) | 1,367 (25%) | 4,218 (29%) | 5,026 (29%) |
| **Deaths within 5 years of COPD diagnosis** | 7,982 (21%) | 709 (13%) | 2,466 (17%) | 4,807 (28%) |

The unlinked group consists of patients who cannot be linked to ONS and have either died within 5 years, or not transferred out of practice within that time. Death data is from CPRD-GOLD, not from ONS. Variable information are presented as either counts (percentages) or median (interquartile range).

similar number of variables as the Charlson co-morbidity index uses (or fewer if possible). However, we wanted to include in addition four variables with known relevance to prognosis of survival in COPD patients: gender, smoking status, body mass index and FEV1% predicted. We used repeated 10-fold cross validation (with five replicates) in the training set of linked patients to compare two models, both of which used information on 21 variables, including age, gender, smoking status, body mass index and FEV1% predicted. These models also included not-recorded indicators for body mass index and FEV1% predicted, as well as quadratic terms for age, body mass index and FEV1% predicted.

The first of the two models additionally included Charlson co-morbidity index, which is derived from information on 16 variables (i.e. diseases). This model was out-performed by iCOPD, which included main effects for the 16 diseases whose variables had the largest absolute log odds ratios in a larger model that included main effects for the 30 co-morbidities with a prevalence >1% in the linked group (S2 Fig).

The iCOPD model had a better overall predictive accuracy and discrimination than models using only basic variables and/or multimorbidity risk scores (i.e. the Charlson co-morbidity index or Cambridge multimorbidity score–S1 and S2 Figs). The model was not noticeably improved by the inclusion of additional co-morbidities, a diagnosis year variable or extra blood tests (eGFR, albumin, c-reactive protein, platelets–S1 Fig).

The iCOPD model was re-fitted to the full training data (80% of practices in the linked group), resulting in the coefficients provided in Table 3 (and S3 Table in machine readable form). In this model, having cancer (odds ratio (OR) 0.44), heart failure (OR 0.44), alcohol problems (OR 0.49) and being older (e.g. ORs 2.0 and 0.47 for ages 59 and 76, respectively, compared to median age of 68) were most negatively associated with survival. In contrast,

**Table 3. Coefficients of the iCOPD model.**

| iCOPD variable | Five-year survival odds ratio (95% CI) |
|---|---|
| Intercept | 6.52 (6.11–6.95) |
| Age in years from 67.7 | 0.918 (0.915–0.920) |
| Age in years from 67.7, squared | 0.999 (0.999–0.999) |
| Body Mass Index in kg/m$^2$ from 26 | 1.04 (1.03–1.04) |
| Body Mass Index in kg/m$^2$ from 26, squared | 0.998 (0.997–0.998) |
| FEV1% predicted from 64.6% | 1.01 (1.01–1.02) |
| FEV1% predicted from 64.6%, squared | 1.00 (1.00–1.00) |
| Female | 1.30 (1.24–1.37) |
| FEV1 not-recorded | 0.582 (0.551–0.616) |
| Never smoker | 1.90 (1.75–2.06) |
| Ex smoker | 1.50 (1.41–1.58) |
| Body Mass Index not recorded | 0.604 (0.559–0.653) |
| Alcohol problems | 0.486 (0.426–0.555) |
| Atrial fibrillation | 0.649 (0.595–0.708) |
| Diabetes | 0.677 (0.629–0.729) |
| Heart failure | 0.444 (0.403–0.489) |
| Inflammatory bowel disease | 0.767 (0.617–0.952) |
| Peripheral vascular disorder | 0.593 (0.538–0.654) |
| Substance abuse | 0.728 (0.596–0.890) |
| Connective tissue disorders | 0.782 (0.707–0.865) |
| Stroke | 0.703 (0.648–0.763) |
| Asthma | 1.26 (1.18–1.35) |
| Cancer | 0.443 (0.399–0.493) |
| Constipation | 0.751 (0.683–0.825) |
| Depression | 0.747 (0.699–0.799) |
| Epilepsy | 0.684 (0.558–0.839) |
| Irritable bowel syndrome | 1.34 (1.21–1.49) |
| Pyschosis/bipolar | 0.662 (0.561–0.781) |

Caution should be taken in the interpretation of these odds ratios, which, while useful for prediction, may be biased. This is especially true for odds ratios for variables with associated not-recorded indicators.

never smoking (OR 1.9) was most positively associated with survival. Not-recorded indicators for FEV1 (OR 0.58) and for BMI (OR 0.60) were negatively associated with survival.

## Validation of models within the held-out test set of English practices

Within the held-out test set of the linked group iCOPD performed well (area under the curve of 0.801, calibration slope of 0.991 and Brier score of 0.139) and comparably to its performance in the training set (Table 4). Actual versus estimated deaths in risk quintiles of the held-out

**Table 4. Actual versus estimated deaths in risk quintiles of the held-out data set for iCOPD.**

| Risk of death quintile | 5-year survival probability range | Dead in 5-years | | Alive in 5-years | |
|---|---|---|---|---|---|
| | | Actual | Estimated | Actual | Estimated |
| 1 (Highest) | [0.00956,0.601] | 1392 (58%) | 1446 (60%) | 1028 (42%) | 974 (40%) |
| 2 | (0.601,0.779] | 732 (30%) | 726 (30%) | 1687 (70%) | 1692 (70%) |
| 3 | (0.779,0.871] | 401 (17%) | 413 (17%) | 2018 (83%) | 2006 (83%) |
| 4 | (0.871,0.929] | 237 (9.8%) | 237 (9.8%) | 2182 (90%) | 2182 (90%) |
| 5 (Lowest) | (0.929,0.991] | 94 (3.9%) | 113 (4.7%) | 2325 (96%) | 2306 (95%) |

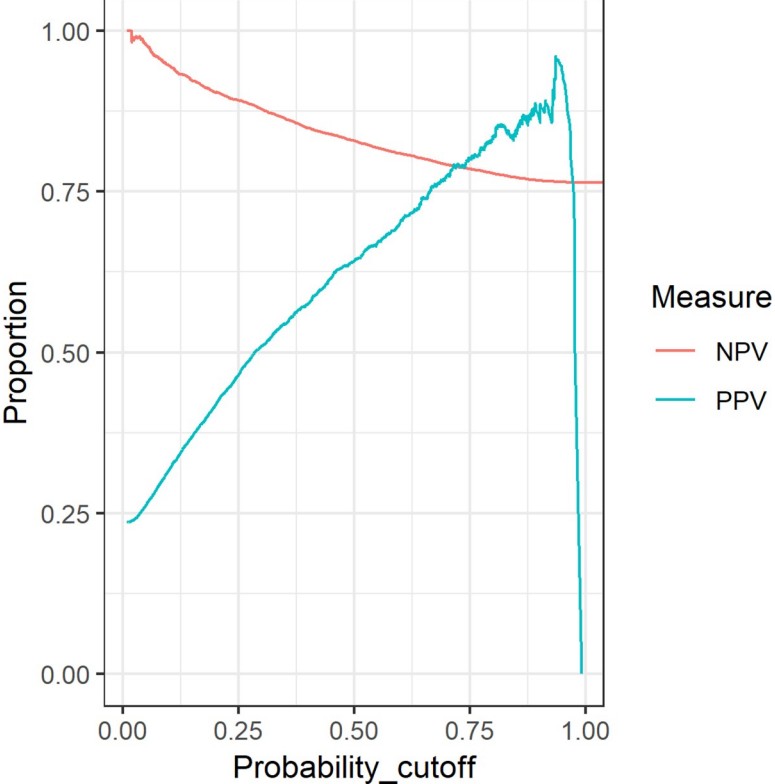

**Fig 2. Positive and negative predictive value (PPV and NPV) for prediction of five-year mortality in the held-out test set across a range of probability cut-offs for the ICP model.**

data set for iCOPD are compared in Table 4. Positive and negative predictive values for both models in the test set across a range of thresholds are given in Fig 2.

## Validation of models within the test set of UK practices

Within the unlinked group, which was not used in model development, iCOPD performed well (area under the curve 0.794, calibration slope 0.978 and Brier 0.134). The performance of iCOPD was comparable between the linked and unlinked patient groups, this was also the case when the unlinked group was stratified by country (Table 5).

The largest difference in performance was seen between the linked and unlinked patients from English practices. However, the performance of iCOPD in unlinked English practices was still acceptable (area under the curve 0.783, calibration slope 0.911 and Brier 0.133).

**Table 5. Comparison of iCOPD validation performance between the linked and unlinked groups, and regions of the UK.**

| Region | Brier score | Calibration slope | AUC |
|---|---|---|---|
| England linked training set (80% of linked group) | 0.139 | 1.00 | 0.797 |
| England linked test set (20% of linked group) | 0.139 | 0.991 | 0.801 |
| All unlinked | 0.134 | 0.978 | 0.794 |
| England unlinked | 0.133 | 0.911 | 0.783 |
| Northern Ireland unlinked | 0.120 | 1.00 | 0.809 |
| Scotland unlinked | 0.139 | 1.01 | 0.790 |
| Wales unlinked | 0.136 | 1.04 | 0.806 |

## Discussion

We have used a large primary care cohort to develop and validate the iCOPD model for the prediction of mortality within 5 years of a COPD diagnosis, using only variables already recorded within health records at the time of diagnosis. iCOPD achieved area under the curves of between 0.783–0.809 and calibration slopes between 0.911–1.04 in validation cohorts from across the UK not used in model development. Being the first models to predict 5-year mortality from the point of COPD diagnosis based only on data already available within health records, there is no direct comparison with existing COPD prognosis scores. However, our models outperformed models using the Charlson co-morbidity index and Cambridge multi-morbidity score risk scores. Importantly, iCOPD had relatively consistent performance between development and validation cohorts. iCOPD is accessible through an online risk calculator (https://skiddle.shinyapps.io/incidentcopdsurvival/).

We used not-recorded indicators for several variables, because it is likely that the fact that data are not recorded within GP records is itself informative of risk. For example, FEV1 data is necessary for COPD diagnosis, and so its absence within GP records at the first recording of COPD is likely to be because patients were diagnosed and tested within secondary care. This could indicate that they are more ill, which is consistent with the negative association of survival with FEV1 not-recorded in GP records.

Limitations of this study include that patients may be misclassified due to undiagnosed co-morbidities, or misdiagnosis of COPD or co-morbidities. However, the use of many relevant co-variates, such as never smoking, will partly account for this. For the unlinked group vital status at five years was unknown for 10% of patients. Therefore, we are encouraged by the similarity of the estimated performance measures between the unlinked and held-out part of the linked group (where vital status was always known). Additionally, due to the observational and prediction-based nature of this study, associations between variables and mortality should not be interpreted causally. As a substantial proportion of COPD patients are on long-term bronchodilators, it is likely that FEV1 measurements are post-bronchodilator. Unfortunately, specific information on whether FEV1 was measured post-bronchodilator is not routinely recorded in UK GP records. Finally, while we have taken care to rigorously assess the predictive model using cross-validation and held-out data, it has not yet been validated using external data, e.g. other GP record systems or other data from non-UK countries. Within the UK consistent clinical and recording practices in GP record systems mean that our models are likely to be relevant [19]. While clinical and recording practice may differ subtly in other European countries, we believe that iCOPD is likely to have utility in these settings (and would like to validate this). In countries, including USA, where diagnosis and management is more often in specialty settings, iCOPD is less likely to have utility.

The focus of our work was on developing a good prediction model, rather than searching for significant associations between individual variables and mortality. However, agreeing with the results of the COTE study [3], we found that cancer was strongly associated with risk of mortality. We see a stronger association between heart failure and death than the COTE study, which may be to do with differences in the populations studied, the data sources (designed study versus primary care records) or the modelling approaches used. Increased risk of mortality in individuals with both heart failure and COPD has previously been found to be associated with intense COPD treatment [20]. Our studies agree that alcohol problems, atrial fibrillation and coronary heart disease are associated with mortality risk. However, we find many more conditions that help to predict mortality in incident COPD patients.

In the future we hope to improve iCOPD with the addition of extra variables (e.g. additional COPD symptoms, exacerbation-like events, severity of co-morbidities, or using less broad co-

morbidity definitions) and the use of longitudinal (i.e. time-varying) data up to the point of diagnosis. We also plan to use to it as the basis of a model that works equally well for both incident and prevalent cases, and dynamically over time. The most important thing to study, however, would be whether iCOPD is useful for clinicians and their COPD patients.

In conclusion, we have developed and validated a model for the prediction of mortality five years after the diagnosis of COPD, providing an online risk calculator. If shown to be helpful, it could be implemented within GP health records, providing prognosis information to GPs automatically using the data that they already collect on their COPD patients.

## Supporting information

**S1 Table. Modelling approaches compared in model development for objective 1, for results see S1 Fig.** The modelling methods were logistic regression, random forests (a popular machine-learning technique) and Cox regression (i.e. Cox proportional hazards). The variable sets were: just basic variables; basic variables and co-morbidity score; and basic variables and co-morbidity indicators. Logistic regression (*glm*) and random forest (*randomForest*) analyse survival as a binary variable: death within five years of COPD diagnosis. Cox regression (*survival*) analyses survival as a time to event outcome, in this case with survival times censored at 5 years after COPD diagnosis. This censoring has been advocated as a way to improve predictions. Logistic regression and Cox regression were performed with ridge penalisation, lasso penalisation or no penalisation, and, when the variable set included co-morbidity indicators, both with and without pairwise interactions between these indicators. The Aalen-Nelson estimator of the baseline hazard was used to make predictions from the fitted Cox regression model. CRP = C-reactive protein. Default settings were used for all methods and nested cross-validation of penalized models was used to choose the penalty parameter (*cv.glmnet*). Co-morbidity indicators and pairwise interactions between co-morbidity indicators were only included in relevant models if they were >1% prevalent, e.g. a pairwise interaction between co-morbidities was only included if at least 1% of patients had both.
(DOCX)

**S2 Table. Recorded prevalence of the 36 co-morbidities from Barnett et al. TIA = Transient Ischemic Attack.**
(DOCX)

**S3 Table. Machine readable table of coefficients of the iCOPD model.**
(CSV)

**S1 Fig. Comparison of modelling approaches detailed in S1 Table, results of 5 repeats of 10-fold cross validation within the training set (80% of the linked group).** Boxplot showing median and interquartile ranges for (a) prediction accuracy (Brier score), (b) discrimination (AUC = Area Under the Curve) and (c) calibration slope of the prediction models. 'B' variables include age, gender, socioeconomic status, smoking status, BMI (value and testing indicator), FEV1% predicted (value and testing indicator). 'CCI' is a single variable (derived from 17 variables), the Charlson Co-morbidity Index. 'CMS' is a single variable (derived from 20 variables), the general Cambridge Multimorbidity Score, which depends on the presence of Barnett co-morbidities. 'C' includes a separate term for each co-morbidity variable. 'C^2' includes main effects and pairwise interactions between each co-morbidity variable. 'All' includes all basic and co-morbidity variables in a non-linear fashion. For (a) and (b) the red dashed line indicates the best median value over all modelling strategies, whereas for (c) it indicates the perfect calibration (slope = 1). CRP = C-reactive protein.
(DOCX)

**S2 Fig. Comparison of 21 variable models with each other, with basic variables only (excluding IMD) and with a larger model.** Results of 5 repeats of 10-fold cross validation within the training set (80% of the linked group). Boxplot showing median and interquartile ranges for (a) prediction accuracy (Brier score), (b) discrimination (AUC = Area Under the Curve) and (c) calibration slope of the prediction models. For (a) and (b) the red dashed line indicates the best median value over all modelling strategies, whereas for (c) it indicates the perfect calibration (slope = 1). 'CCI' is a single variable, the Charlson Co-morbidity Index. 'Basic–IMD (B-I)' is a model using age, gender, smoking status, body mass index (BMI) and Forced Expiratory Volume in 1-second (FEV1) % predicted, as well as quadratic terms for age, BMI and FEV1% predicted, and not-recorded indicators for BMI and FEV1% predicted. '(B-I) + CCI' adds the Charlson Co-morbidity Index (CCI) to the previous model, this index is calculated using data on 16 co-morbidities. Our 21 variable model uses adds 16 co-morbidity main effects to the '(B-I)' model, as such it uses the same number of variables as the '(B-I) + CCI' model.
(DOCX)

**S1 File. Study protocol approved by Clinical Practice Research Datalink Independent Scientific Advisory Committee.**
(DOC)

**S2 File. TRIPOD reporting guidelines checklist.**
(DOCX)

## Acknowledgments

We acknowledge CPRD @ University of Cambridge for developing and sharing disease definitions, and Silvia Mendonica and Duncan Edwards (University of Cambridge) in particular for advice on implementing these and the Cambridge multimorbidity score. We would like to thank peer reviewers whose comments improved our manuscript. This study is based in part on data from the Clinical Practice Research Datalink obtained under licence from the UK Medicines and Healthcare products Regulatory Agency. The data is provided by patients and collected by the NHS as part of their care and support. ONS is the provider of ONS mortality data used in this study. ONS and HES data copyright © (2018), re-used with the permission of The Health & Social Care Information Centre. All rights reserved. The interpretation and conclusions contained in this study are those of the author/s alone.

## Author Contributions

**Conceptualization:** Steven J. Kiddle, Jennifer K. Quint.

**Data curation:** Steven J. Kiddle, Hannah R. Whittaker.

**Formal analysis:** Steven J. Kiddle.

**Funding acquisition:** Steven J. Kiddle.

**Investigation:** Steven J. Kiddle.

**Methodology:** Steven J. Kiddle, Shaun R. Seaman.

**Project administration:** Steven J. Kiddle.

**Software:** Steven J. Kiddle.

**Supervision:** Jennifer K. Quint.

**Validation:** Steven J. Kiddle.

**Visualization:** Steven J. Kiddle.

**Writing – original draft:** Steven J. Kiddle.

**Writing – review & editing:** Steven J. Kiddle, Shaun R. Seaman, Jennifer K. Quint.

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
