## [Decision Letter · Decision Letter 0]

28 May 2020

PONE-D-20-12164

Prediction of five-year mortality after COPD diagnosis using primary care records

PLOS ONE

Dear Dr. Kiddle,

Thank you for submitting your manuscript to PLOS ONE. After careful consideration, we feel that it has merit but does not fully meet PLOS ONE’s publication criteria as it currently stands. Therefore, we invite you to submit a revised version of the manuscript that addresses the points raised during the review process.

We look forward to receiving your revised manuscript.

Kind regards,

Konstantinos Kostikas, M.D., Ph.D.

Academic Editor

PLOS ONE

Journal Requirements:

'Dr. Kiddle reports grants from Medical Research Council, during the conduct of the study; personal fees from Roche Diagnostics and DIADEM, outside the submitted work. After completing this work, but before manuscript submission Dr. Kiddle became an employee of AstraZeneca. Ms. Whittaker reports grants from GlaxoSmithKline, during the conduct of the study. Dr. Seaman has nothing to disclose. Dr. Quint reports grants from MRC, grants from The Health Foundation, grants from BLF, grants and personal fees from GSK, grants and personal fees from BI, grants and personal fees from Insmed, grants and personal fees from AZ, personal fees from Chiesi, personal fees from Teva, outside the submitted work.'

5. Your ethics statement must appear in the Methods section of your manuscript. If your ethics statement is written in any section besides the Methods, please move it to the Methods section and delete it from any other section. Please also ensure that your ethics statement is included in your manuscript, as the ethics section of your online submission will not be published alongside your manuscript.

Reviewers' comments:

Reviewer's Responses to Questions

**Comments to the Author**

1. Is the manuscript technically sound, and do the data support the conclusions?

Reviewer #1: Yes

Reviewer #2: Yes

2. Has the statistical analysis been performed appropriately and rigorously? 

Reviewer #1: Yes

Reviewer #2: Yes

3. Have the authors made all data underlying the findings in their manuscript fully available?

Reviewer #1: Yes

Reviewer #2: Yes

4. Is the manuscript presented in an intelligible fashion and written in standard English?

Reviewer #1: Yes

Reviewer #2: Yes

5. Review Comments to the Author

Reviewer #1: This is an interesting study aiming to propose a new model for mortality prediction in newly diagnosed COPD patients as a tool for general practicioners. While there are several composite scores or individual predictors used to assess the mortality risk, the novelty of this model is that it includes basic informations about the patient and severity of COPD that are likely available at the GP level. However, I would mention a few issues with this approach:

1. It is well known that the mortality correlates well with the health status in general and the level of COPD symptoms in particular. The current model does not include any data on the symptom level, although this should be routinely collected to all incident COPD patients.

2. Exacerbations are events with a major impact on the disease evolution and mortality risk. While I acknowledge the difficulty in identifying such episodes in a previously undiagnosed COPD patient, "exacerbation-like" events could probably be identified in the patient's records from the previous year. This information could be essential for future risk assessment and I suggest it should be included in the model.

3. The criteria for differential diagnosis between asthma and COPD in the current study was only historical. However the presence of asthma was correlated in the study with better vital prognosis. Therefore, it would be important to ensure a proper differential diagnosis between the 2 disease using also the lung function data, since an old asthma may look like a COPD, but the prognosis may be different.

4. FEV1 is a good predictor of mortality risk at a population level, but not at individual level. Therefore a single measurement of this parameter may not give accurate information on the mortality risk.

5. While the information provided by this model on the 5 years mortality risk is certainly useful, I believe that a shorter interval (e.g. 3 years or 1 year) would be more helpful in informing the therapeutic strategy. Did the authors consider a shorter period of time for the modelling of the death risk?

6. Finally, the model would probably fit to those countries with a UK-like primary care system. The performance of the model proposed by the authors should therefore be validated accross different health care systems, including countries where the diagnosis and management of the COPD patients is primarily carried at a secondary or tertiary care level.

Reviewer #2: Congratulations to the authors for their original and interesting work.

However, I would like to make some comments and raise a few questions that, in my opinion, have to be answered before approval.

Minor Revisions

1. In Line 74: add … to make informed decisions about....

2. In Line 363 in the discussion session the authors comment that FEV1 data absence may be meaningful by itself, suggesting that, among others, it may indicate that COPD diagnosis is likely to be confirmed within secondary care and consequently it may indicate a more severe disease. However, as this suggestion is speculative, authors should emphasize that this could lead in misdiagnosis and have an impact in the strength of their results.

3. The authors should make a comment about their finding of a stronger association between heart failure and death compared to cancer and provide similar findings in other studies, if any.

4. In the discussion section

6. PLOS authors have the option to publish the peer review history of their article (what does this mean?). If published, this will include your full peer review and any attached files.

Reviewer #1: Yes: Stefan Marian Frent

Reviewer #2: No

---

## [Author Response · Author response to Decision Letter 0]

23 Jun 2020

An easier to read colour coded version of this included in the files for review.

---

Dear reviewers and editors,

Many thanks for your feedback which has helped us to improve our manuscript. Our detailed responses are below

Reviewer #1: 

This is an interesting study aiming to propose a new model for mortality prediction in newly diagnosed COPD patients as a tool for general practicioners. While there are several composite scores or individual predictors used to assess the mortality risk, the novelty of this model is that it includes basic informations about the patient and severity of COPD that are likely available at the GP level. However, I would mention a few issues with this approach:

1. It is well known that the mortality correlates well with the health status in general and the level of COPD symptoms in particular. The current model does not include any data on the symptom level, although this should be routinely collected to all incident COPD patients.

2. Exacerbations are events with a major impact on the disease evolution and mortality risk. While I acknowledge the difficulty in identifying such episodes in a previously undiagnosed COPD patient, "exacerbation-like" events could probably be identified in the patient's records from the previous year. This information could be essential for future risk assessment and I suggest it should be included in the model.

We thanks the reviewer for their advice. We include Forced Expiratory Volume in 1-second (FEV1) in our model, which is the most commonly recorded of the symptoms of COPD at the point of diagnosis. Most symptoms per se tend to be recorded in the free text rather than as coded data. As we cannot access the free text from the GP record, we are unable to include that information. We now discuss in more details variables we could add to the model in the future (such as additional symptoms and "exacerbation-like" events) in the discussion. From discussion “In the future we hope to improve iCOPD with the addition of extra variables (e.g. additional COPD symptoms, exacerbation-like events, severity of co-morbidities, or using less broad co-morbidity definitions)”

3. The criteria for differential diagnosis between asthma and COPD in the current study was only historical. However the presence of asthma was correlated in the study with better vital prognosis. Therefore, it would be important to ensure a proper differential diagnosis between the 2 disease using also the lung function data, since an old asthma may look like a COPD, but the prognosis may be different.

We thank the reviewer for highlighting this. The association of asthma with better prognosis is not unique to patients with a COPD diagnosis, as it has also been seen in the general population in the Cambridge Multimorbidity Score paper. We handle the potential for misdiagnosis between asthma and COPD in two ways: (1) as mentioned by reference to historical data (in a process that we have validated to have high positive predictive value by contacting GPs in a separate study), and (2) by including asthma and lung function in the model. The only additional variable that could be used to aid the separation of these groups would be Forced Vital Capacity, but this is too sparsely recorded in GP records to be useful for this purpose

4. FEV1 is a good predictor of mortality risk at a population level, but not at individual level. Therefore a single measurement of this parameter may not give accurate information on the mortality risk.

We agree, and we list time-varying data (e.g. longitudinal FEV1) in our future work (see below), but it is not common for multiple measures of FEV1 to be available at the point of COPD diagnosis. We would like to point out that the performance of our model is already good and validates well, but there is always scope for improvements in the future.

From discussion “In the future we hope to improve iCOPD with the addition of extra variables (e.g. additional COPD symptoms, exacerbation-like events, severity of co-morbidities, or using less broad co-morbidity definitions) and the use of longitudinal (i.e. time-varying) data up to the point of diagnosis. We also plan to use to it as the basis of a model that works equally well for both incident and prevalent cases, and dynamically over time”

5. While the information provided by this model on the 5 years mortality risk is certainly useful, I believe that a shorter interval (e.g. 3 years or 1 year) would be more helpful in informing the therapeutic strategy. Did the authors consider a shorter period of time for the modelling of the death risk?

For our next piece of work, using longitudinal historical data, we plan to generate predictions at multiple time horizons. 

6. Finally, the model would probably fit to those countries with a UK-like primary care system. The performance of the model proposed by the authors should therefore be validated accross different health care systems, including countries where the diagnosis and management of the COPD patients is primarily carried at a secondary or tertiary care level.

We couldn’t agree more, and have commented in our original discussion:

“While clinical and recording practice may differ subtly in other European countries, we believe that iCOPD is likely to have utility in these settings (and would like to validate this). In countries, including USA, where diagnosis and management is more often in specialty settings, iCOPD is less likely to have utility.”

We hope to be able to validate this ourselves, but also release model coefficients to allow others to validate it in data they have access to. 

Reviewer #2: Congratulations to the authors for their original and interesting work.

However, I would like to make some comments and raise a few questions that, in my opinion, have to be answered before approval.

Minor Revisions

1. In Line 74: add … to make informed decisions about....

Edit made as suggested

2. In Line 363 in the discussion session the authors comment that FEV1 data absence may be meaningful by itself, suggesting that, among others, it may indicate that COPD diagnosis is likely to be confirmed within secondary care and consequently it may indicate a more severe disease. However, as this suggestion is speculative, authors should emphasize that this could lead in misdiagnosis and have an impact in the strength of their results.

We thank the reviewer for their suggestion. While we do not explicitly link missing FEV1 to the risk of COPD misdiagnosis, we do list patient misclassification due to misdiagnosis of COPD as a study limitation (see below). We discuss elsewhere the steps we have taken to reduce bias due to this, such as including a ‘never smoker’ indicator in our model, as COPD is less likely in these individuals, and efforts to reduce misdiagnosis between asthma and COPD based on historical data and inclusion of FEV1 and an asthma indicator in our model.

“We used not-recorded indicators for several variables, because it is likely that the fact that data are not recorded within GP records is itself informative of risk. For example, FEV1 data is necessary for COPD diagnosis, and so its absence within GP records at the first recording of COPD is likely to be because patients were diagnosed and tested within secondary care. This could indicate that they are more ill, which is consistent with the negative association of survival with FEV1 not-recorded in GP records.

Limitations of this study include that patients may be misclassified due to undiagnosed co-morbidities, or misdiagnosis of COPD or co-morbidities. However, the use of many relevant co-variates, such as never smoking, will partly account for this.”

3. The authors should make a comment about their finding of a stronger association between heart failure and death compared to cancer and provide similar findings in other studies, if any.

We do not see a stronger association of heart failure to death than cancer to death in our study, rather that these associations are closer in strength than in the COTE study. We are not aware of other papers showing this but we now remind the reader that looking for specific associations was not the focus of our paper, to reduce the risk that this is over-interpreted, and provide potential explanations for the discrepancy. To back-up the importance of co-morbid heart failure and COPD we cite an additional paper:

“The focus of our work was on developing a good prediction model, rather than searching for significant associations between individual variables and mortality. However, agreeing with the results of the COTE study [3], we found that cancer was strongly associated with risk of mortality. We see a stronger association between heart failure and death than the COTE study, which may be to do with differences in the populations studied, the data sources (designed study versus primary care records) or the modelling approaches used. Increased risk of mortality in individuals with both heart failure and COPD has previously been found to be associated with intense COPD treatment [20].” 

Lawson CA, Mamas MA, Jones PW, et al. Association of Medication Intensity and Stages of Airflow Limitation With the Risk of Hospitalization or Death in Patients With Heart Failure and Chronic Obstructive Pulmonary Disease. JAMA Netw Open. 2018;1(8):e185489. doi:10.1001/jamanetworkopen.2018.5489

---

## [Editor Report · Decision Letter 1]

29 Jun 2020

Prediction of five-year mortality after COPD diagnosis using primary care records

PONE-D-20-12164R1

Dear Dr. Kiddle,

We’re pleased to inform you that your manuscript has been judged scientifically suitable for publication and will be formally accepted for publication once it meets all outstanding technical requirements.

Kind regards,

Konstantinos Kostikas, M.D., Ph.D.

Academic Editor

PLOS ONE
---

## [Editor Report · Acceptance letter]

1 Jul 2020

PONE-D-20-12164R1 

Prediction of five-year mortality after COPD diagnosis using primary care records 

Dear Dr. Kiddle:

I'm pleased to inform you that your manuscript has been deemed suitable for publication in PLOS ONE. Congratulations! Your manuscript is now with our production department. 

Kind regards, 

on behalf of

Dr. Konstantinos Kostikas 

Academic Editor

PLOS ONE